# In Vivo Antidepressant-Like Effect Assessment of Two *Aloysia* Species in Mice and LCMS Chemical Characterization of Ethanol Extract

**DOI:** 10.3390/molecules27227828

**Published:** 2022-11-13

**Authors:** Teresa Taboada, Nelson L. Alvarenga, Antonia K. Galeano, Wilfrido J. Arrúa, Miguel A. Campuzano-Bublitz, María L. Kennedy

**Affiliations:** 1Departamento de Farmacología, Facultad de Ciencias Químicas, Universidad Nacional de Asunción, Campus Una, San Lorenzo 2169, Paraguay; 2Departamento de Fitoquímica, Facultad de Ciencias Químicas, Universidad Nacional de Asunción, Campus Una, San Lorenzo 2169, Paraguay

**Keywords:** *Aloysia virgata* var. *platyphylla*, *Aloysia gratissima* var. *gratissima*, depression, tail suspension test, forced swimming test

## Abstract

Medicinal plants belonging to the Verbenaceae family demonstrated antidepressant effects in preclinical studies. Depression is one of the largest contributors to the global health burden of all countries. Plants from the *Aloysia* genus are traditionally used for affective disorders, and some of them have proven anxiolytic and antidepressant activity. The aim of this work was to evaluate the antidepressant effect of the ethanolic extract of *Aloysia gratissima* var. *gratissima* (Agg) and *Aloysia virgata* var. *platyphylla* (Avp) in mice. A tail suspension test (TST) and forced swimming test (FST) were conducted after three doses in a period of 24 h and after 7 days of treatment. Imipramine was used as an antidepressant drug. The main results demonstrated that Agg extract reduced the immobility time in mice treated orally for 7 consecutive days when compared to the control group (reduced by about 77%, imipramine 70%). Animals treated with three doses of Avp in a 24-h period had reduced immobility time in the FST (60%), and after 7 days of treatment the reduction was greater (Avp 50, 100, and 200 about 85%; Avp 400, 96.5%; *p* < 0.0001, imipramine, 77%). LCMS analysis showed the presence of verbascoside, hoffmaniaketone, and hoffmaniaketone acetate in both, *A. virgata* var. *platyphylla* and *A. gratissima* var *gratissima*. The flavonoids nepetin and 6-hydroxyluteolin were also found in Agg. Both tested extracts demonstrated promising antidepressant-like activity in mice.

## 1. Introduction

Depression is a common mental disorder and represents a global mental health concern [1]. Depressive disorders are characterized by sadness, loss of interest, poor concentration, low self-worth, feelings of guilt and tiredness, disturbed sleep and/or appetite, and so on [2]. Before 2020, mental disorders were a leading cause of the global health-related burden, with depressive disorder being the leading contributor to this burden [3]. In Paraguay, it is estimated that around 5.23% of the population suffers from depressive disorders, ranking fourth in the Americas, surpassed only by the United States, Brazil, and Cuba [4].

Depression increases the risk of severe cardiovascular diseases. In addition, it can be considered a risk factor for chronic diseases and, on the other hand, depression might arise as a result of physical limitations and disabilities that lead to chronic diseases [5].

Pharmacological therapy for depressive disorders uses tricyclic antidepressants, monoamine oxidase inhibitors, selective serotonin reuptake inhibitors, serotonin and norepinephrine reuptake inhibitors, norepinephrine and dopamine reuptake inhibitors, and serotonin antagonist and reuptake inhibitors [6,7]. Some patients receive substantial benefits, while others have less or no benefit from the treatment [8]. Moreover, these medications might cause several adverse effects, such as sedation, anticholinergic effects, weight gain, seizures, impotence, anxiety, postural hypotension, and cardiac dysrhythmias [9].

Many medicinal plants from indigenous cultures are used for the treatment of multiple conditions. This knowledge has been transmitted from generation to generation, by oral communication, and become part of the culture of our countries. Many species have demonstrated pharmacological potential for use as an antidepressant in humans, such as *Crocus sativus* L. (Iridaceae), *Humulus lupulus* L. (Cannabaceae), *Melissa officinalis* L. (Lamiaceae), and *Hypericum perforatum* L. (Hypericaceae) [10,11,12,13,14,15]. Some South American medicinal plants that act on the central nervous system have more citations than others in the literature, such as Lamiaceae, Asteraceae, and Verbenaceae [13]. In the Verbenaceae family, one of the species, *Aloysia polystachya*, has been shown to exert an antidepressant effect in mice, rats, and fish [16,17,18]. *Aloysia gratissima*, another species of the same family, has shown central activity, and the aqueous extract has shown an antidepressant effect in mice after acute treatment [19,20]. *Aloysia gratissima* var. *gratissima* methanolic extract was effective as an anxiolytic [21].

*Aloysia virgata* var. *platyphylla* grows in Brazil, Bolivia, Argentina, and Paraguay [22]. It is used as an anti-catarrhal, antirheumatic, diaphoretic, stimulant, stomachic, and emollient [23]. The anxiolytic effect of *Aloysia virgata* var. *platyphylla* in elevated plus maze and hole board tests has been reported, as well as its affinity for the GABA_A_ receptor in the 3-[H]flunitrazepam binding assay [24]. Currently, there is no report on the antidepressant activity of this plant.

Considering the background and the high prevalence of depression worldwide, interest arises in evaluating the antidepressant effect of the ethanolic extract of *Aloysia gratissima* var. *gratissima* and *Aloysia virgata* var. *platyphylla* after acute and sub-chronic treatment. The present research focuses on evaluation of the antidepressant activity of these two *Aloysia* species in mice by determining acute antidepressant activity after three doses in a period of 24 h and after 7 days of treatment via tail suspension and forced swimming tests.

## 2. Results

### 2.1. Aloysia Gratissima var. Gratissima

#### 2.1.1. Effect of Oral Treatment with *A. Gratissima* var. Gratissima in the Tail Suspension Test

Following treatment with three doses of *A. gratissima* var. *gratissima* (Agg) extract in a period of 24 h, no statistically significant decrease was observed in the immobility time of the animals compared to the group treated with the vehicle (Veh). A statistically significant decrease in immobility time was observed in animals treated with imipramine, the reference antidepressant used in this trial, when compared to animals treated with the vehicle (40.5 ± 7.5 s; 110.1 ± 21.6 s, respectively, *p* < 0.0001, Figure 1).

A statistically significant reduction in immobility time was observed in animals treated for 7 days with Agg 50 and animals treated with imipramine when compared to the vehicle group (86.1 ± 29.7 s *p* < 0. 01; 40.4 ± 25.2 s *p* < 0.0001; and 130.3 ± 22.3 s, respectively, Figure 2).

#### 2.1.2. Effect of Oral Treatment with A. Gratissima var. Gratissima in the Forced Swimming Test

The animals received three doses of *A. gratissima* var. *gratissima* (Agg) in a period of 24 h (50, 100, 200, and 400 mg/kg). A statistically significant reduction in immobility time was observed in Agg 200 and imipramine-treated animals when compared to animals treated with the vehicle (64.7 ± 32.6 s, *p* < 0.05; 37.0 ± 20.6 s, *p* < 0.001; and 115.1 ± 21.9 s, respectively, Figure 3).

However, after being treated for a period of 7 consecutive days with the same doses, a statistically significant reduction in the immobility time of the animals treated was verified with these doses of Agg extract, in addition to those that were treated with imipramine, when compared to the control group (Agg 50: 57.0 ± 32.8 s; Agg 100: 60.3 ± 29.4 s; Agg 200: 44.2 ± 17.0 s; Agg 400: 49.5 ± 20 0.8 s; IM 40.5 ± 21.7 s; Veh (control) 179.3 ± 37.6 s; *p* < 0.0001, Figure 4). Apparently, this reduction in immobility time is not dose-dependent, since approximately the same percentage of reduction (about 77%) was observed with all the tested doses and the percentage is similar to that obtained with imipramine (70%). A statistically significant increase is observed in swimming time, which is inversely proportional to the time of immobility. There is no statistically significant increase in the climbing behavior of the animals after treatment in a 24-h period or after 7 days, and there was no difference in the imipramine-treated animals compared to the vehicle group (Table 1).

### 2.2. Aloysia Virgata var. Platyphylla

#### 2.2.1. Effect of Oral Treatment with *A. Virgata* var. *Platyphylla* in the Tail Suspension Test

After per os treatment with three doses in a 24-h period of the Avp extract, no statistically significant decrease was observed in the immobility time of the animals in relation to the control group (Veh, Figure 5). The immobility time of the animals treated with the reference antidepressant, imipramine, decreased significantly compared to the control group (40.5 ± 7.5 s and 110.1 ± 21.6 s, respectively; *p* < 0.0001, Figure 6). On the other hand, a statistically significant reduction in immobility time was observed in animals treated for 7 days with Avp 50, Avp 100, and imipramine when compared to the control group (Avp 50: 84.3 ± 34.8 s, *p* < 0.05; Avp 100: 83.5 ± 28.5 s, *p* < 0.05; Im: 32.7 ± 16.1 s, *p* < 0.0001; Veh: 134.2 ± 21.7 s).

#### 2.2.2. Effect of Oral Treatment with A. Virgata var. platyphylla in the Forced Swimming Test

The animals that received three doses of the extract of *A. virgata* var. *platyphylla* administered orally in a 24 h period, as well as those treated with imipramine, had reduced immobility time, and the difference compared to the control group was significant (Avp 50: 45.3 ± 29.6 s, *p* < 0.001; Avp 100: 53.7 ± 42.0 s, *p* < 0.01; Avp 200: 43.5 ± 13.5 s, *p* < 0.001; Avp 400: 50.7 ± 12.6 s, *p* < 0.01; Im: 31, 3 ± 15.5 s, *p* < 0.0001; Veh: 118.8 ± 21.5 s, Figure 7). The reduction in immobility time is apparently not dose-dependent, since approximately the same percentage reduction (60%) was observed with all doses tested; imipramine reduced the immobility time by up to 73%.

Similarly, after oral treatment with the extract for 7 days, a statistically significant reduction (*p* < 0.0001) in immobility time was observed with all doses tested and with imipramine when compared to the vehicle-treated group (Avp 50: 22.2 ± 19.6 s; Avp 100: 23.8 ± 9.2 s; Avp 200: 30.0 ± 17.3 s; Avp 400: 6.2 ± 4.8 s; Im: 40.5 ± 21.7 s; Veh: 179.3 ± 37.6 s, Figure 8). In this case, there is an apparent dose–response relationship, since the reduction percentage is greater after treatment with Avp 400 (96.5%) than that obtained after treatment with Avp 50, 100, or 200 (about 85%) compared to the control group. This reduction is even greater than that obtained with imipramine (77%). No difference was observed in the climbing behavior of the animals, neither with the treatment of three doses in a 24-h period, nor after 7 days (Table 2).

#### 2.2.3. Compound Identification by LCMS

The compounds present in the extracts were identified by LCMS, which compared the molecular ions obtained with those of the compounds previously identified in the *Aloysia* genus. Verbascoside ([M − H]^−^ 623.9), hoffmaniaketone ([M + H]^+^ 321.2), hoffmaniaketone acetate, plus an isomer ([M + H]^+^ 363.2) were identified from *Aloysia virgata* var. *platyphylla* (Figure 9). These compounds were previously described in the same specie [25]. 

The following were identified from *Aloysia gratissima* var. *gratissima* (Figure 10): verbascoside, hoffmaniaketone, hoffmaniaketone acetate, plus the flavonoids nepetin ([M − H]^−^ 315.3) and 6-hydroxyluteolin ([M − H]^−^ 301.2), already described in other *Aloysia* species [26,27].

## 3. Discussion

Immobility time in the FST and TST is an index for measuring antidepressant-like activity. Antidepressant drugs reduce immobility time and increase swimming behavior. Our results allowed us to prove that *Aloysia gratissima* var. *gratissima* and *Aloysia virgata* var. *platyphylla* were able to induce a statistically significant reduction in the immobility time of the animals in the tail suspension test and in the forced swimming test in both treatment modalities (acute treatment and treatment for 7 days). Immobility time reduction induced by imipramine demonstrates that the methods are valid when identifying potentially useful candidates to treat depression. [28,29,30].

Reduction of immobility time in the FST and TST has been previously reported in mice treated with a single dose of the aqueous extract of *Aloysia gratissima* [20]. In this study, a statistically significant decrease in immobility time in the tail suspension test was observed only in the group of animals treated with Agg 50 for 7 days (*p* < 0.05). Hoffmaniaketone, a compound identified in Agg, acts on the GABA_A_ receptor [24], and because of that probably induces a slight increase in immobility time in mice after acute treatment.

In the forced swimming test, after treatment with three doses in 24 h of Agg 200, immobility time was also significantly reduced (*p* < 0.05). When compared to the vehicle, after 7 days of treatment, a reduction in immobility time was observed with all doses of Agg tested (*p* < 0.0001). These times are much shorter than those obtained in the 24-h test and those obtained by Zeni et al. for *A. gratissima* [20]. It is interesting to note that the reductions with Agg and imipramine were almost the same, around 70%; these results do not show a dose-dependent relationship. Another Verbenaceae, *Aloysia polystachya*, showed that both acute and 7-day administration decreased immobility time in the forced swimming test in mice [17].

Regarding climbing behavior in the forced swimming test, no statistically significant difference was observed between the groups treated with Agg. Nevertheless, a statistically significant increase in swimming time was observed in animals treated with all doses of the extract, as expected. These results agree with the reported effect on sleep time induced by barbiturates [21], thus indicating that Agg exerts its antidepressant effect without exerting a stimulating effect on the CNS [31].

The results reported by Zeni et al. after the administration of a single dose of *A. gratissima* indicated that one of the probable mechanisms of action involves the non-selective inhibition of serotonin and norepinephrine reuptake (as with imipramine), results that agree with the climbing and swimming behavior time reported in this study [19,20].

The difference observed between the immobility time of the animals treated with the aqueous extract of *A. gratissima*, in the tail suspension test reported by Zeni et al., and those obtained in this study, with the ethanolic extract of *A. gratissima* var *gratissima*, could be due to differences in the extraction procedure, the plant we have studied (a sub-species), or the harvest season (summer). In addition, there are differences in the environmental and feeding conditions of the Swiss albino mice used from the FCQ animal facility and in the methodology used [19,20]. Moroni et al. described a complex of four varieties of *A. gratissima*, which probably contributes to differences in the qualitative and quantitative composition of secondary metabolites and therefore to the effect observed in terms of immobility time [32].

*A. gratissima* may interact with serotonergic, dopaminergic, and noradrenergic systems according to the TST results of Zeni et al. [19]. However, it is known that the tail suspensions test is not effective enough to determine substances that exert an antidepressant effect through other signaling pathways such as the GABAergic system [33]. Taking this into account, and the differences between our result with the ethanolic extract of *A. gratissima* var. *gratissima* and the anxiolytic effect reported [21], it could indicate, as a possible mechanism of action of the antidepressant effect, a set of interactions with other receptors not studied by Zeni et al., for which the tail suspension method is not sensitive enough, such as the inhibition of MAO or interaction with the GABA_A_ receptor [33,34,35].

Avp did not alter the immobility time of animals treated with three doses in 24 h in the tail suspension test. This could be due to the previously reported anxiolytic effect of two diterpenes isolated from this species, which agrees with the results reported, that is, the anxiolytic effect of the methanolic extract of *A. virgata* var. *platyphylla* in mice [24,36]. Additionally, one of the diterpenes reported here showed moderate affinity to the benzodiazepine binding site in the GABA_A_ receptor [24].

In this study, it was shown that all doses of the extract induced a statistically significant decrease in immobility time in the forced swimming test after treatment for 24 h and for 7 days. As with *A. gratissima* var. *gratissima*, this statistically significant reduction in immobility time occurred without an increase in climbing behavior, though a statistically significant increase in swimming behavior was observed. Avp has demonstrated that it is not a CNS stimulant [36].

An advantage of differentiating the types of behaviors in animals is the possibility of distinguishing the CNS-stimulating effect from the antidepressant effect and providing an idea about the mechanism of action, since those antidepressant drugs whose mechanism of action involves the non-selective inhibition of reuptake of serotonin and norepinephrine decrease the time that animals display immobility behavior and increase the time that animals display swimming behavior, without modifying climbing behavior [29,30]. Based on the results observed, it could be inferred that Agg and Avp exert an antidepressant effect without exerting a stimulating effect on the CNS by a mechanism of action that could involve the same molecular targets involved in the mechanism of action of imipramine.

A current theory about one of the causes of symptoms of depression postulates decreased gamma-aminobutyric acid in the occipital and prefrontal cortex in magnetic resonance spectroscopy studies in acutely depressed patients, in addition to the theory of a decreased release of monoamines in the synaptic cleft [37]. Therefore, both the interaction with the benzodiazepine binding site in the GABA_A_ receptor and the increase in the immobility time observed in the tail suspension test could explain one of the mechanisms by which we observed the antidepressant effect in the animals treated with the different doses of *A. gratissima* var. *gratissima* and *A. virgata* var. *plathyphylla*.

Anxiolytic drugs such as diazepam exert their effect through this mechanism, and this drug increases immobility time in the tail suspension test [24,33,35]. The increase in immobility time when treating with Agg and Avp in the TST could be related to the anxiolytic effect previously described [21,36], whose effect is very similar to that observed with diazepam. Moreover, Avp showed better performance on both tests. This may be due to the amount of active metabolite present in Avp, the compound or compounds that remain without the full characterization of being responsible for the antidepressant effect. This should be addressed in subsequent studies. Verbascoside (acteoside), identified as the main constituent of *Aloysia polystachya*, was considered responsible for the antidepressant effect of this plant [16]. In addition, verbascoside has been proven to have a neuroprotective effect [38] and a modulatory effect on monoamine oxidase activity [39]. This compound, identified in both extracts, may be responsible for the antidepressant-like effect of *A. gratissima var. gratissima* and *Aloysia virgata var. platyphylla*, and the mechanism could involve MAO inhibition. Additionally, the presence of the diterpenes hoffmaniaketone and hoffmaniaketone acetate is reported here, with the former showing the ability to bind to the GABA_A_ receptor [24]; therefore, anxiolytic activity probably has a role in the mechanism of antidepressant activity. Finally, it is important to mention that nepetin and ten other flavonoids were isolated from a fraction with antidepressant activity, determined by a bioguided assay, of *Inula japonica* [40].

Molecular analysis should be carried out to explore the mechanism involved in the antidepressant effect observed. Additionally, isolation and quantification of the secondary metabolites identified in both extracts is required to make a more precise comparison of the antidepressant efficacy of each plant.

## 4. Materials and Methods

### 4.1. Plant Material and Extraction

The aerial parts of *Aloysia gratissima* var. *gratissima* (Gillies & Hook. ex Hook.) Tronc. and *Aloysia virgata* var. *platyphylla* (Briquet.) Moldenke were collected from the Garden of Acclimatization of Native and Medicinal Plants of the Facultad de Ciencias Químicas (FCQ), Universidad Nacional de Asunción (San Lorenzo, Paraguay), and identified by researchers from the Department of Botany-FCQ. After being dried at room temperature for 48 h in a ventilated environment and protected from the sun, they were reduced to a fine powder. Subsequently, 200 g of the powder was extracted with 1000 mL of ethanol for 15 min in an ultrasonic bath at 50 °C (three times). The extract was filtered off and the residue was further extracted by reflux (three times). Solvent was removed on a rotary evaporator. The extracts obtained were kept in a desiccator until their use in biological assays.

### 4.2. Animals

Male Swiss albino mice weighing between 25 and 35 g were used. The animals were kept in a 12-h light/dark cycle, climatized at 23–25 °C with a relative humidity of 40–70%, and given free access to food and water. The night before the experiments, the food was removed so that the animals were in a fasting state, though they still had free access to water unless otherwise specified. All animals were subjected to previous evaluations to rule out those showing abnormal behaviors that could interfere with the results.

### 4.3. Reagents and Chemicals

Imipramine hydrochloride was obtained from Wako Pure Chemical Industries Ltd. (Osaka, Japan) and sodium chloride was procured from the Sigma Chemical Company (St. Louis, MO, USA). Ethanol, Tween 80, and propylene glycol for pharmaceutical use were locally purchased. All chemicals were of analytical grade. All the reagents used for the LCMS analysis were of LCMS grade.

### 4.4. Tail Suspension Test

Groups of adult male Swiss albino mice (*n* = 6) were tested to evaluate possible antidepressant actions. The tail suspension test (TST) involves suspending mice above the ground by their tails, for a period of 6 min, the total immobility time for each animal was recorded from the end of the second minute. Treatment with antidepressant drugs decreases the immobility time of antidepressant-treated animals when compared to the immobility time of vehicle-treated animals in the tail suspension test [28]. In this study, the effect of different doses of the extract was analyzed after the administration of three doses in a period of 24 h and after seven days of administration. The groups of animals received doses of the vehicle (tween 80 at 10%; 0.1 mL/10 g of weight, orally), *A. gratissima* var. *gratissima* (Agg) or *A. virgata* var. *platyphylla* (Avp) extract (50, 100, 200, and 400 mg/kg, orally), or Imipramine (Im 32 mg/kg, intraperitoneal route). In the first case (three doses in a period of 24 h), this treatment was administered 24, 18, and 1 h before being individually subjected to the tail suspension test. In the second case (seven days), the animals were treated for 7 consecutive days, and one hour after the last administration they were individually subjected to the same tail suspension procedure [28]. The doses of the extract were selected based on previously reported acute toxicity results [16].

### 4.5. Forced Swimming Test (FST) in Mice

The animals treated as previously detailed (after being submitted to the tail suspension test) were observed in the forced swimming test. They were individually introduced into a transparent cylinder (25 cm high and 15 cm in diameter) containing 12.5 cm of water maintained between 21 and 24 °C. Each 6-min session was videotaped for subsequent analysis. The immobility, climbing, and swimming time were then measured (in seconds), starting at the end of the second minute [24,28]. Mice were considered to be immobile when they made the minimum movements necessary to keep their head above water. Climbing behavior is defined as movements of the forelegs upward along the side of the swimming cylinder. Swimming behavior is defined as the usually horizontal movement through the swimming cylinder that usually includes crossings into another quadrant. In all cases, the water was changed after every other trial and the cylinder was cleaned with 10% ethanol solution. Antidepressant drugs decrease immobility time in the forced swim test for treated animals when compared to vehicle-treated animals. Additionally, the increase in climbing or swimming behavior could indicate other effects on the CNS or the mechanism of antidepressant action [29,30].

### 4.6. LCMS Analysis

For this purpose, the samples were dissolved in LC-MS grade acetonitrile (Merck KGaA, Darmstadt, Germany) at 10 mg/mL and filtered with a 0.22 µm Nylon sample filter (Microclar, Buenos Aires, Argentina). Subsequently, they were injected into a liquid chromatograph coupled to a tandem mass spectrometer (Xevo-TQD, Waters Corporation, Milford, MA, USA) with an electrospray ionization (ESI) source under the following conditions: Waters BEH C18 column (150 mm × 2.1 mm × 1.7 µm, Waters, Milford, MA, USA), flow rate of 0.3 mL/min, gradient elution using a mixture of water (with 0.1% formic acid and 10 mM ammonium formate, eluent A) with acetonitrile (with 0.1% formic acid, eluent B), LC-MS grade reagents (Merck KGaA, Darmstadt, Germany). The gradient conditions were as follows: start with 100% A, maintain those conditions for 1 min, decrease A to 0% at 7.75 min and maintain those conditions until 10 min, increase A to 50% at 15 min, increase again to 95% at 18 min, and maintain those conditions until the end (20 min). Mass spectrometer conditions were positive and negative mode ionization, ramped cone voltage from 18.80 to 30 V, capillary voltage of 3.20 (+) and 3.90 (−) kV, respectively, source temperature of 150 °C, desolvation temperature of 500 °C, and desolvation gas flow of 1000 L/h.

The molecular mass obtained experimentally was compared to the molecular masses of the compounds described in the literature regarding the genus *Aloysia*. For this purpose, a bibliographic search was performed in databases such as PubChem, Lotus, and Google scholar using the keywords: *Aloysia*/Chemical composition/Phytochemical composition.

### 4.7. Statistical Analysis

The data obtained in the different study groups were expressed as means ± standard deviation. Statistical analysis was performed using GraphPad Prism 7.0 software (La Jolla California, USA) using one-way ANOVA followed by Dunnett’s multiple comparison test. A level of *p* < 0.05 was considered statistically significant. GraphPad Prism version 7.0 for Windows, GraphPad Software, (www.graphpad.com, accessed on 19 October 2022).

### 4.8. Ethical Issues

As this was an experimental study with laboratory animals, they were considered to be biological reagents and work was thus carried out in accordance with the standards established by the European Community Ethics Commission. The handling of the animals was carried out by standardized procedures and the basic rule to follow was that “all treated animals must be euthanized”. The protocol was approved by the Research Ethics Committee of the Facultad de Ciencias Químicas, UNA (Code CEI 718/2021). The minimum number of animals and the shortest duration of observation required to obtain consistent data were used. Each animal was used once [41].

## 5. Conclusions

The antidepressant activity of the ethanolic extracts of *Aloysia gratissima* var. *gratissima* (Agg) and *Aloysia virgata* var. *platyphylla* (Avp) was evaluated in tail suspension and forced swimming tests in animals treated with multiple doses in 24 h or animals treated for 7 days. A. *gratissima* var. *gratissima* and *A. virgata* var. *platyphylla* decreased the immobility time of the mice treated in the forced swimming test in both treatment modalities. The effect was much greater after 7 days of treatment. The best antidepressant-like activity was seen with Avp. It was concluded that the ethanolic extracts of *A. gratissima* var. *gratissima* and *A. virgata* var. *platyphylla* demonstrated an antidepressant effect in mice. Verbascoside, hoffmaniaketone, and nepetin may be partially responsible for the antidepressant activity in the samples studied.

## Figures and Tables

**Figure 1 molecules-27-07828-f001:**
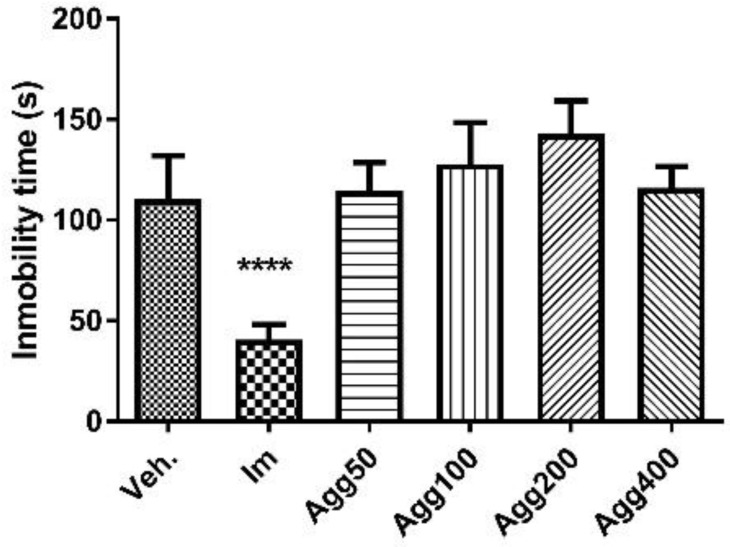
Immobility time (s) in the tail suspension test for the groups (*n* = 6) treated orally with three doses in a 24-h period with *Aloysia gratissima* var. *gratissima* (Agg). Im.: imipramine. Veh: tween 80, 10%. The data correspond to the mean ± SD. One-way ANOVA and Dunnett’s post hoc test were performed. **** *p* < 0.0001.

**Figure 2 molecules-27-07828-f002:**
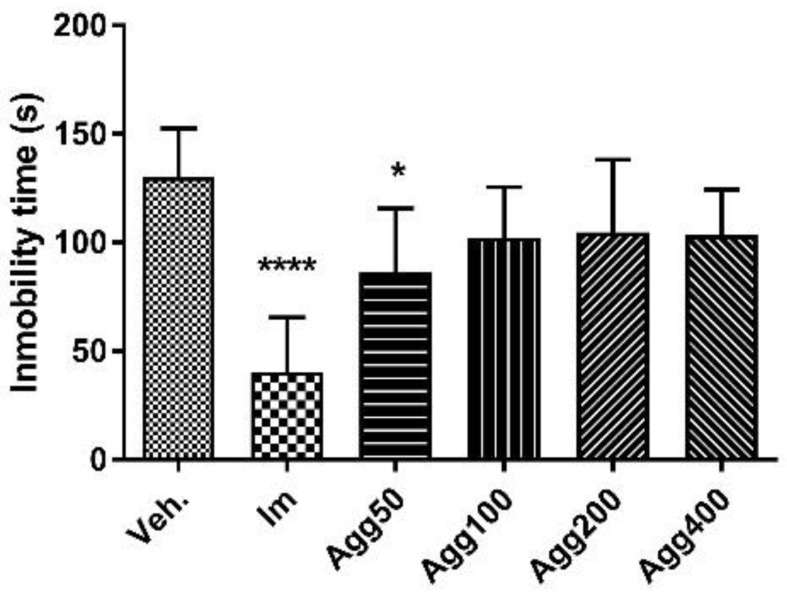
Immobility time (s) in the tail suspension test for the groups (*n* = 6) treated orally for 7 days with *Aloysia gratissima* var. *gratissima* (Agg). Im.: imipramine. Veh: tween 80, 10%. The data correspond to the mean ± SD. One-way ANOVA and Dunnett’s post hoc test were performed. * *p* < 0.05; **** *p* < 0.0001.

**Figure 3 molecules-27-07828-f003:**
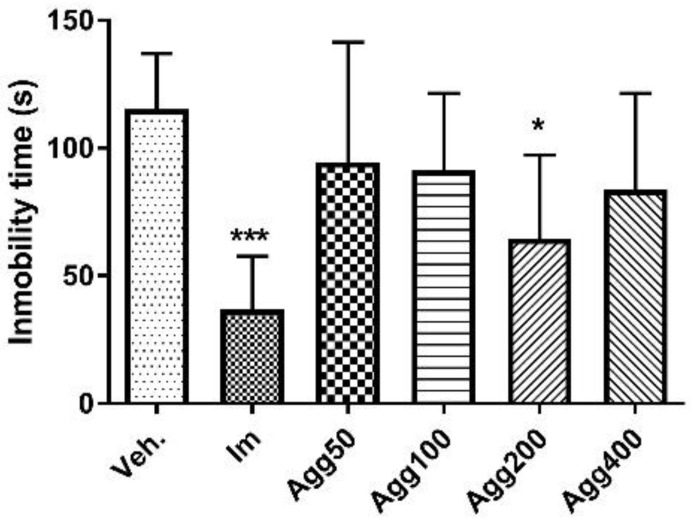
Immobility time (s) in the forced swimming test for the groups (*n* = 6) treated orally with three doses in a 24-h period of *Aloysia gratissima* var. *gratissima* (Agg). Im.: imipramine. Veh: tween 80, 10%. The data correspond to the mean ± SD. One-way ANOVA and Dunnett’s post hoc test were performed. * *p* < 0.05; *** *p* < 0.001.

**Figure 4 molecules-27-07828-f004:**
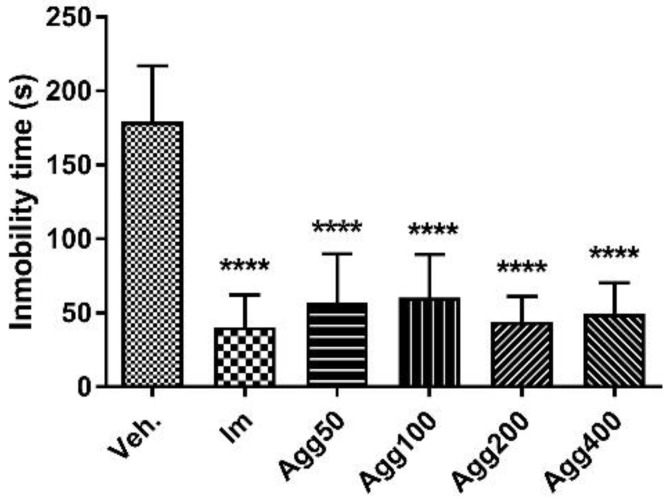
Immobility time (s) in the forced swimming test for the groups (*n* = 6) treated orally for 7 days with *Aloysia gratissima* var. *gratissima* (Agg). Im.: imipramine. Veh: tween 80, 10%. The data correspond to the mean ± SD. One-way ANOVA and Dunnett’s post hoc test were performed. **** *p* < 0.0001.

**Figure 5 molecules-27-07828-f005:**
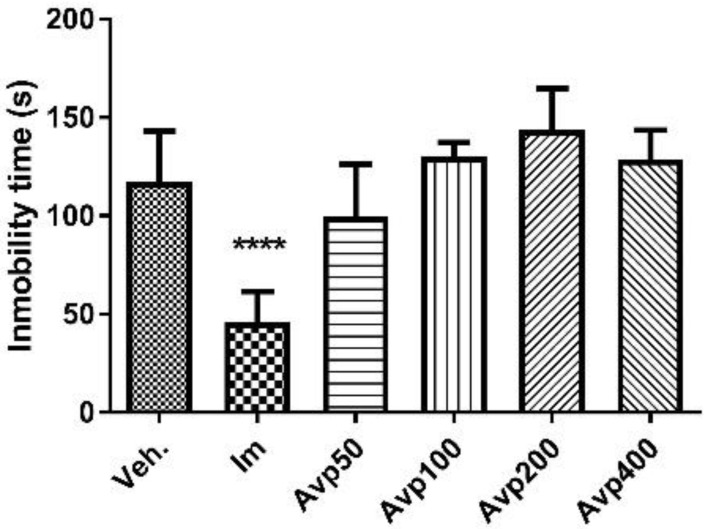
Immobility time (s) in the tail suspension test for the groups (*n* = 6) treated orally with three doses in a 24-h period of *Aloysia virgata* var. *platyphylla* (Avp). Im.: imipramine. Veh: tween 80, 10%. The data correspond to the mean ± SD. One-way ANOVA and Dunnett’s post hoc test were performed. **** *p* < 0.0001.

**Figure 6 molecules-27-07828-f006:**
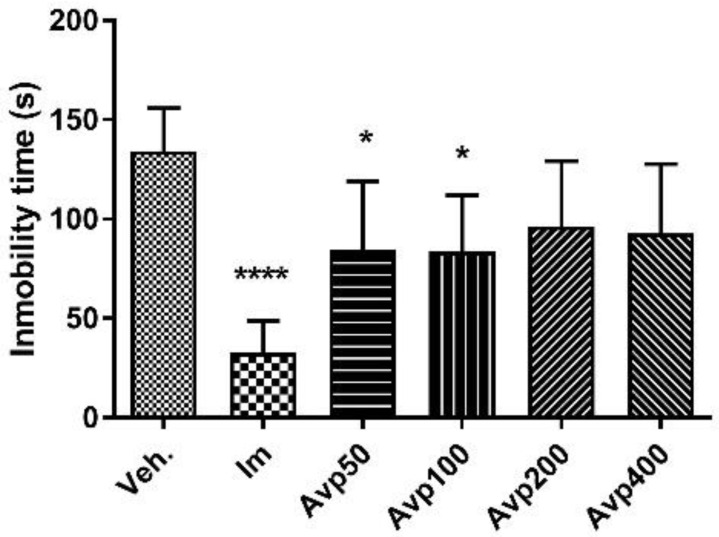
Immobility time (s) in the tail suspension test for the groups (*n* = 6) treated orally for 7 days with *Aloysia virgata* var. *platyphylla* (Avp). Im.: imipramine. Veh: tween 80, 10%. The data correspond to the mean ± SD. One-way ANOVA and Dunnett’s post hoc test were performed. * *p* < 0.05; **** *p* < 0.0001.

**Figure 7 molecules-27-07828-f007:**
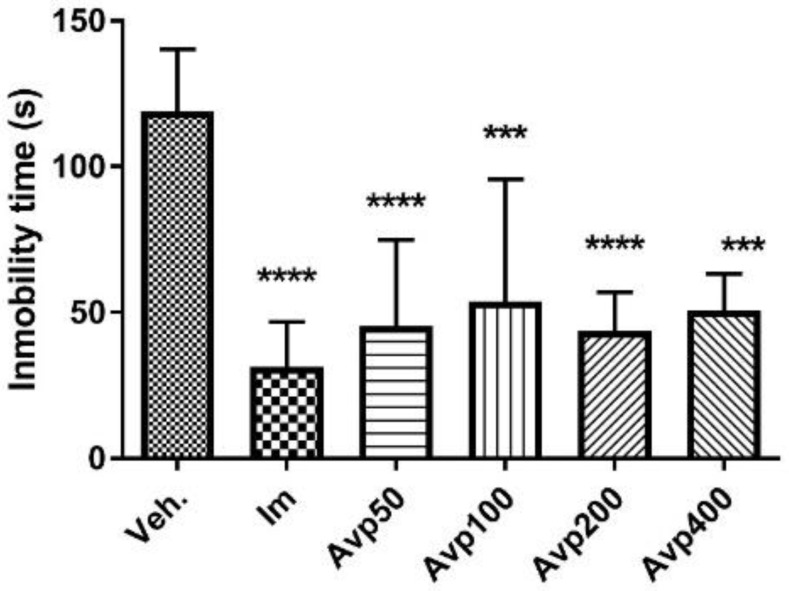
Immobility time (s) in the forced swimming test for the groups (*n* = 6) treated orally with three doses in a 24-h period of *Aloysia virgata* var. *platyphylla* (Avp). Im.: imipramine. Veh: tween 80, 10%. The data correspond to the mean ± SD. One-way ANOVA and Dunnett’s post hoc test were performed. *** *p* < 0.001; **** *p* < 0.0001.

**Figure 8 molecules-27-07828-f008:**
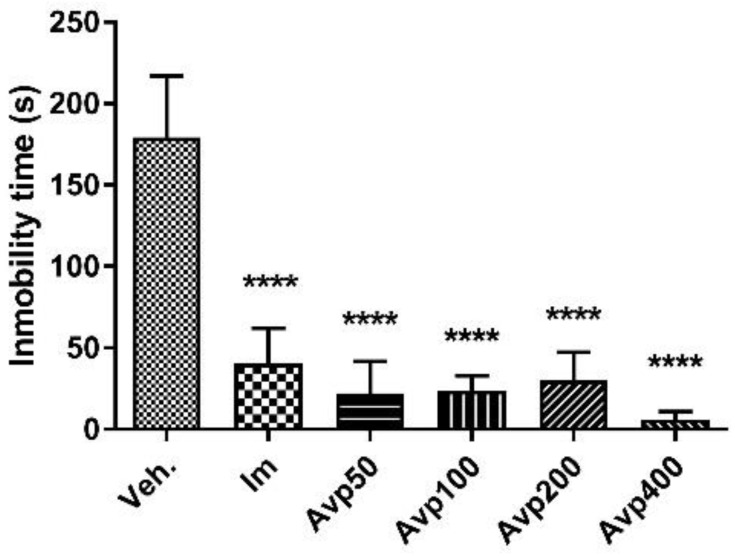
Immobility time (s) in the forced swimming test for the groups (*n* = 6) treated orally for 7 days with *Aloysia virgata* var. *platyphylla* (Avp). Im.: imipramine. Veh: tween 80, 10%. The data correspond to the mean ± SD. One-way ANOVA and Dunnett’s post hoc test were performed. **** *p* < 0.0001.

**Figure 9 molecules-27-07828-f009:**
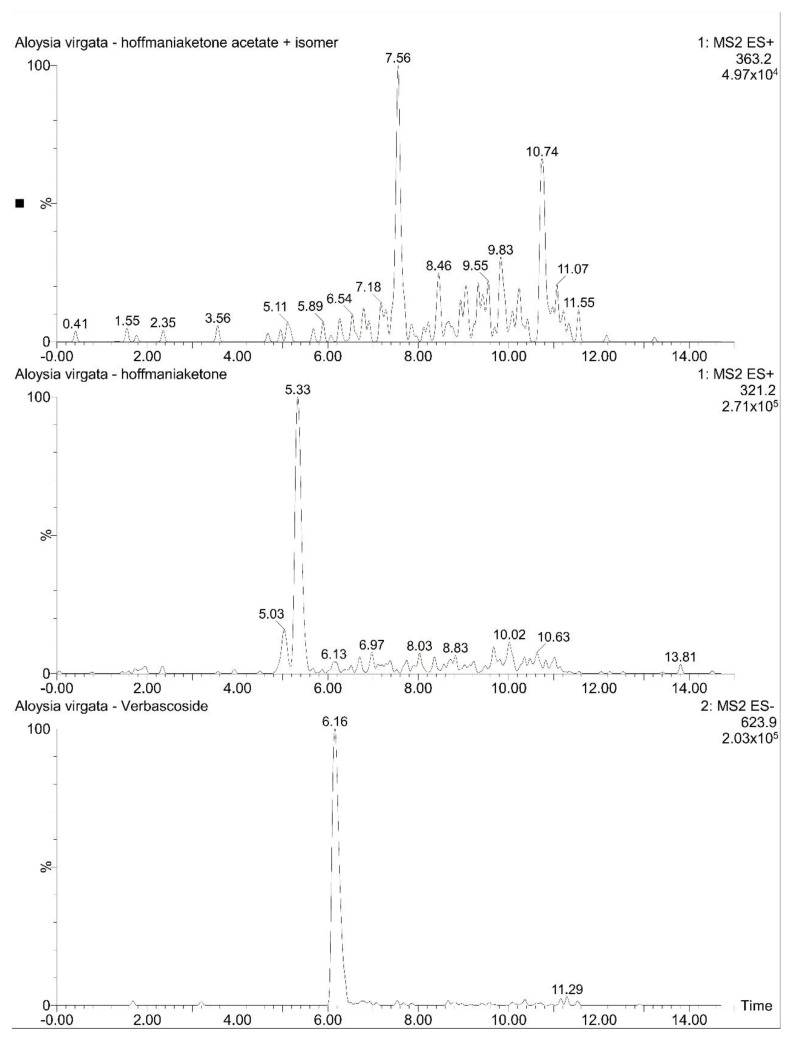
Extracted ESI-LCMS ion chromatograms of compounds from *Aloysia virgata* var. *platyphylla*.

**Figure 10 molecules-27-07828-f010:**
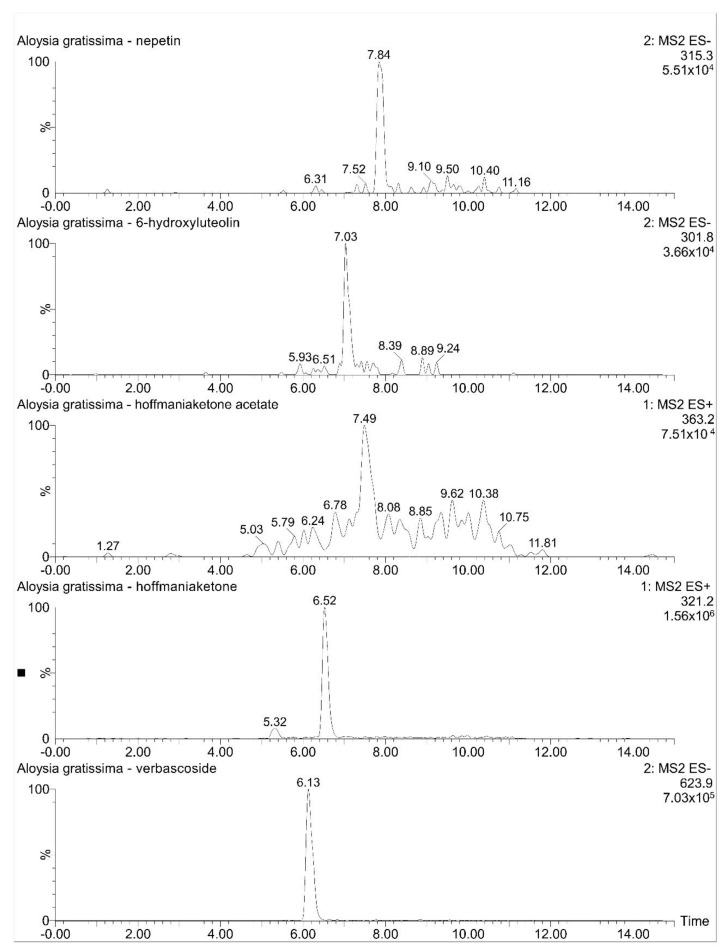
Extracted ESI-LCMS ion chromatograms from *Aloysia gratissima* var *gratissima*.

**Table 1 molecules-27-07828-t001:** Climbing and swimming behavior time of mice (*n* = 6) after treatment with three doses in a 24-h period of *Aloysia gratissima* var. *gratissima*.

	Three Doses in 24 h		7 Days of Treatment	
Treatment	Swimming (s)	Climbing (s)	Swimming (s)	Climbing (s)
Veh.	120.8 ± 21.4	1.5 ± 2.8	54.3 ± 38.9	6.3 ± 5.9
Im.	197.2 ± 19.0	2.0 ± 3.5	191.8 ± 22.5	7.7 ± 8.7
Agg 50	149.5 ± 44.9	4.7 ± 4.8	174.8 ± 30.1	8.2 ± 7.6
Agg 100	153.2 ± 25.4	3.5 ± 6.2	176.3 ± 32.3	3.3 ± 6.2
Agg 200	180.3 ± 29.8	1.3 ± 1.3	192.7 ± 16.7	3.2 ± 2.8
Agg 400	162.7 ± 36.3	1.3 ± 1.5	182.8 ± 19.1	7.7 ± 3.9

**Table 2 molecules-27-07828-t002:** Climbing and swimming behavior time of mice (*n* = 6) after treatment with three doses in a 24-h period of *Aloysia virgata* var. *platyphylla*.

	Three Doses in 24 h		7 Days of Treatment	
Treatment	Swimming (s)	Climbing (s)	Swimming (s)	Climbing (s)
Veh.	120.3 ± 20.8	1.5 ± 2.8	54.3 ± 38.9	6.3 ± 5.9
Im.	197.2 ± 19.0	2.0 ± 3.5	191.8 ± 22.5	7.7 ± 8.7
Avp 50	193.0 ± 29.7	1.7 ± 0.0	214.2 ± 21.4	3.7 ± 3.5
Avp 100	185.5 ± 42.6	0.8 ± 0.0	212.0 ± 10.0	4.2 ± 3.9
Avp 200	195.7 ± 14.2	1.7 ± 0.0	195.2 ± 13.3	14.8 ± 7.9
Avp 400	188.8 ± 12.4	0.3 ± 0.0	222.0 ± 15.3	12.2 ± 12.6

## Data Availability

All data generated and analyzed are included within this research article.

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
