# Peer review of "In Vivo Antidepressant-Like Effect Assessment of Two Aloysia Species in Mice and LCMS Chemical Characterization of Ethanol Extract"

_molecules, 2022, doi:10.3390/molecules27227828_

Round 1

Reviewer 1 Report

Title: Antidepressant-like effect of two Aloysia species in mice and LCMS chemical characterization

The authors have written the manuscript in a very sequential and scientific way. This manuscript is well-designed and well-described and covered all necessary parameters. There are many concerns/flaws and areas which should be improved before publishing it.  

1.      Title of the study is somewhat different from the core objective of the study, it should be modify according to the main results of this work.

2.      The abstract should be started with mentioning background of Aloysia plant whether it has any medicinal properties reported in the literature.

3.      Abstract should be amended with some numerical values from the key results of the study.

4.      Line 24, 25 the abstract should be concluded on a suitable elaboration sentences that attract the readers related to this work. 

5.      In introduction section: line 30 to 41, in description of depression, it is suggested to rephrase the para and elaborate more by supporting more relevant updated research articles;

https://doi.org/10.3390/biomedicines10102385, https://doi.org/10.3390/molecules27175462.

6.      Line 42 to 57 in introduction section; the author stick to mentioned the citation of work already published in a specific region which is not suitable for readers working on these species in other part of the world, modify the section with a broader independent literature.

7.      Results: the author should add the figures, tables, schemes in their respective sections of the study.

8.      Most of the figures have poor lay out with deviation of the symbol (*) from respective graph bars.

9.      The author did not mention in this study, how much was the sample size? In each case of  (figure and table) where animals were used, sample size should be indicated.

10.   The author performed Dunnets post hoc tests using One-Way ANOVA for analyzing the results. Is each set of data was in normal distribution? Which type of test was used for normal distribution of each data set, it should be mention in the caption of each figure.   

11.   The discussion section should be initiated with para showing significances of the study. Moreover, the author divides the discussion in to different section, in my opinion the subsections should be removed and the results should discuss paragraphs.

12.   Line 248 typos mistake; as evidenced should be started with As evidenced.

13.   In methods: section: doses of AVP 50, AVP 100, AVP 200, AVP 400 were used. How the author determines that these doses were suitable for this work, did the author perform any acute toxicity study before going to conduct TST and FST for the extract? The author should mention it.

14.   Conclusion of this study is too lengthy; it should be concise on the basis of results and its opinion for future endurance.

15.   The authors should provide limitation of the study.

16.   In most cases the references are too old, It should be updated from recent studies.

17.   The article should be revised thoroughly for grammatical mistakes.

18.   In my opinion, it should be thoroughly revised addressing all the concerns before publishing it. 

Author Response

Response to Reviewer 1 Comments

Thanks for your valuable feedback on our manuscript. We are grateful for the insightful comments. We incorporated several changes to reflect all the suggestions.

The authors have written the manuscript in a very sequential and scientific way. This manuscript is well-designed and well-described and covered all necessary parameters. There are many concerns/flaws and areas which should be improved before publishing it.

We appreciate this comment.

  1. Title of the study is somewhat different from the core objective of the study; it should be modified according to the main results of this work.

We modified the title

  1. The abstract should be started with mentioning background of Aloysia plant whether it has any medicinal properties reported in the literature.

Changed as reviewer suggested, see revised document

  1. Abstract should be amended with some numerical values from the key results of the study.

Changed as reviewer suggested, see revised document

  1. Line 24, 25 the abstract should be concluded on a suitable elaboration sentence that attract the readers related to this work. 

Changed as reviewer suggested, see revised document

  1. In introduction section: line 30 to 41, in description of depression, it is suggested to rephrase the para and elaborate more by supporting more relevant updated research articles; https://doi.org/10.3390/biomedicines10102385, https://doi.org/10.3390/molecules27175462

Changed as reviewer suggested, see revised document, re-elaborated, and more updated references were added

  1. Line 42 to 57 in introduction section; the author stick to mentioned the citation of work already published in a specific region which is not suitable for readers working on these species in other part of the world, modify the section with a broader independent literature.

Changed as reviewer suggested, see revised document, plants with proved antidepressant effect were added

  1. Results: the author should add the figures, tables, schemes in their respective sections of the study.

The document is according to template (not changed)

  1. Most of the figures have poor lay out with deviation of the symbol (*) from respective graph bars.

All figures were adjusted

  1. The author did not mention in this study, how much was the sample size? In each case of  (figure and table) where animals were used, sample size should be indicated.

This information was added in figures and tables labels

  1. The author performed Dunnets post hoc tests using One-Way ANOVA for analyzing the results. Is each set of data was in normal distribution? Which type of test was used for normal distribution of each data set, it should be mention in the caption of each figure.   

We use done way ANOVA; the statistical model assumes the normality of the data as a primary requirement to perform the programmed statistical calculations.

  1. The discussion section should be initiated with para showing significances of the study. Moreover, the author divides the discussion into different section, in my opinion the subsections should be removed, and the results should discuss paragraphs.

Changed as reviewer asked

  1. Line 248 typos mistake; as evidenced should be started with As evidenced.

Corrected

  1. In methods: section: doses of AVP 50, AVP 100, AVP 200, AVP 400 were used. How the author determines that these doses were suitable for this work, did the author perform any acute toxicity study before going to conduct TST and FST for the extract? The author should mention it.

This information is already mentioned in “Tail suspension test section”, highlighted for reviewer

  1. Conclusion of this study is too lengthy; it should be concise on the basis of results and its opinion for future endurance.

The conclusion was shortened and written in a more concise way

  1. The authors should provide limitation of the study.

Added at the end of discussion

  1. In most cases the references are too old, It should be updated from recent studies.

Several information and references were changed

  1. The article should be revised thoroughly for grammatical mistakes.

The manuscript has been reviewed by a proficient English speaker with experience in academic proofreading. All suggested changes were included.

  1. In my opinion, it should be thoroughly revised addressing all the concerns before publishing it. 

The document was completely revised. We hope we have answered all your comments appropriately.

With best regards,

Reviewer 2 Report

The authors, Taboada et al., investigated the ethanolic extract of two plants from the Verbenaceae family. Previous studies suggested that some of the compounds present in these extracts may exhibit antidepressant activity. This lays a sound background and justification for the study. The authors proposed an animal (mice) model and subjected the animals to two behavioral tests for assessing the antidepressant activity. The extracts were tested against a positive control - imipramine.

The study has merit, and most of the results are well described and interpreted. Below please find comments that may help the authors to correct and improve their manuscript:

1. Have the authors assessed the extracts administered in the study quantitatively and not only qualitatively? Did the active ingredient amounts differ between the extracts? Without standardization, it is challenging to assess what was the effective dose.

2. The authors identified only three active ingredients in Agg and five in Avp. Can only these few exert antidepressant activity?

3. Figures 9 and 10 - please, annotate the panels with the compound names.

4. Discussion - this section requires significant changes. The test results between the extracts did not differ much. Therefore, instead of a somewhat repetitive interpretation of Agg and Avp, the Reviewer suggests a joint discussion of both extracts.

5. The surprising difference in the test outcomes (a significant effect in the swim test [immobility time] vs. lack of it in the tail suspension test) should be discussed in more in-depth. 

6. How can the authors explain that despite the lack of terpenes in Agg (and the authors discuss that these terpenes may have moderate anxiolytic effects similarly to diazepam), the immobility time in the tail suspension test is similar to the control?

7. The authors mention that the tail suspension test is not sensitive enough to detect GABAergic pathways of antidepressant activity. At the same time, they hypothesize that Agg and Avp ingredients may be involved in other pathways, such as MAO inhibition. If so, why did neither acute nor prolonged (except 7-day administration of Avp) treatment significantly decrease the immobility time?

8. The discussion would benefit from a more head-to-head comparison of both extracts.

9. How was the LC-MS analytical method developed? How was it validated? How did the authors confirm the compounds' identities besides comparing them with literature? Are these compounds available as pure standards?

Author Response

Response to Reviewer 2 Comments

  1. Have the authors assessed the extracts administered in the study quantitatively and not only qualitatively? Did the active ingredient amounts differ between the extracts? Without standardization, it is challenging to assess what was the effective dose.

A quantitative analysis of the components of the extract has not been performed. Certainly, the standardization of each of the extracts will be a crucial point when they will eventually be tested in other trials, or in clinical trials. In this phase, we have demonstrated the effective doses, all doses, after 7 days of treatment. The % reduction in immobility after 7 days of treatment, with both extracts, was higher than Imipramine. Therefore, the effectiveness of the tested doses has been demonstrated, regardless of the amount of active component. Note that the active components are not determined. Also, it does not mean that only those compounds are the active components.

  1. The authors identified only three active ingredients in Agg and five in Avp. Can only these few exert antidepressant activity?

it does not mean that only those compounds are the active components, but as we mentioned in discussion, some of them have been tested and demonstrated antidepressant-like activity.

  1. Figures 9 and 10 - please, annotate the panels with the compound names.

Corrected, compound names were added in panels

  1. Discussion - this section requires significant changes. The test results between the extracts did not differ much. Therefore, instead of a somewhat repetitive interpretation of Agg and Avp, the Reviewer suggests a joint discussion of both extracts.

All discussion was revised according to this point of view.

  1. The surprising difference in the test outcomes (a significant effect in the swim test [immobility time] vs. lack of it in the tail suspension test) should be discussed in more in-depth. 

All discussion was revised according to this point of view.

  1. How can the authors explain that despite the lack of terpenes in Agg (and the authors discuss that these terpenes may have moderate anxiolytic effects similarly to diazepam), the immobility time in the tail suspension test is similar to the control?

All discussion was revised according to this point of view.

  1. The authors mention that the tail suspension test is not sensitive enough to detect GABAergic pathways of antidepressant activity. At the same time, they hypothesize that Agg and Avp ingredients may be involved in other pathways, such as MAO inhibition. If so, why did neither acute nor prolonged (except 7-day administration of Avp) treatment significantly decrease the immobility time?

All discussion was revised according to this point of view.

  1. The discussion would benefit from a more head-to-head comparison of both extracts.

All discussion was revised according to this point of view.

  1. How was the LC-MS analytical method developed? How was it validated? How did the authors confirm the compounds' identities besides comparing them with literature? Are these compounds available as pure standards?

The LCMS method is an in-house method of the Phytochemistry Department. As it is a qualitative method, validation is not required, as a validation procedure needs standards in order to evaluate linearity, precision, recovery, and so on. No other procedure except the comparison of the masses with those of compounds already described in the same species or other species of the same genus was performed. We do not have standards for these compounds available.

We hope we have answered all your comments appropriately.

With best regards,

Round 2

Reviewer 1 Report

Agree with authors. All the suggested comments has been endorsed. No further comments from my side. accept for publication

Reviewer 2 Report

All the Reviewer's comments have been addressed.